# The Impact of Digital Economy on the Efficiency of Green Financial Investment in China’s Provinces

**DOI:** 10.3390/ijerph19148884

**Published:** 2022-07-21

**Authors:** Xiangyu Tian, Yuanxi Zhang, Guohua Qu

**Affiliations:** 1School of Accounting, Shanxi University of Finance and Economics, Taiyuan 030006, China; 19931022@sxufe.edu.cn; 2School of Public Administration, Shanxi University of Finance and Economics, Taiyuan 030006, China; 3School of Management Science and Engineering, Shanxi University of Finance and Economics, Taiyuan 030006, China

**Keywords:** digital economy, green finance, high-quality development, green audit

## Abstract

The global digital operation of finance has accelerated, and the transformation and upgrading of the financial industry has been fully empowered by digital technology, which has promoted the development of traditional financial industry toward green finance. Accelerating the pace of China’s entry into the digital economy era has given the green financial industry new opportunities in its digital transformation. Therefore, the research reported in this paper selects provincial panel data and discusses the impact efficiency of the digital economy on green financial investment in China by static panel OLS and the threshold model method, and constructs a threshold model with regional industry scale and green financial audit as threshold variables. These steps are used to analyze the nonlinear characteristics of digital economy and green financial efficiency. The results show that the digital economy can improve the overall efficiency of China’s green finance, and there are two threshold effects with regional industry scale as the threshold variable and one threshold effect with green financial audit support as the threshold variable. The results show that the development of a digital economy improved the investment efficiency of green finance in all provinces of China. In addition, through our research, we found that the application of the digital economy in green finance can reduce the imbalance of regional economic development. China should also strengthen the supervision of green auditing to promote the development of new green financial formats.

## 1. Introduction

In recent years, economic globalization has continuously promoted the development of foreign trade and foreign investment. Additionally, the development of a digital economy has been conducive to solving many structural contradictions in economic development. Recently, Awanetal et al. [1] stated that the financial industry has strengthened the combination with information technology, and a comprehensive model covering different types of enterprises–digital finance has been launched. It is believed that digital finance overcomes the shortcomings of traditional finance, and greatly improves inclusiveness by providing convenient services and lower barriers to entry. The development of a digital economy is beneficial for solving many structural contradictions in economic development and promoting high-quality economic development. This has indicated the direction and provided reference for the development of China’s digital economy. China is facing the challenges of environment and development. How to promote the growth of green finance by reducing energy consumption, protecting the environment, and maintaining economic development has become one of the problems that China needs to solve at present. China’s economy is currently undergoing transformation and upgrading, and the emergence of the digital economy has provided us with new ideas for the development of green finance. The deep integration of the digital economy and traditional green finance creates a new format and model, which is conducive to promoting high-quality growth of economic investment, enhancing the new impetus of China’s economic development and providing strong kinetic energy for economic and social development. China’s digital economy has maintained a high-speed growth trend and has become an important engine to promote economic growth. In 2019, the size of China’s digital economy reached 35.8 trillion, CNY accounting for 36.2% of GDP. Thus, whether the digital economy can effectively improve the investment efficiency of green finance is the core content of this paper.

It has been found that the continuous application and development of a digital economy also promotes the transformation and expansion of the green financial industry. China’s information and communication technology is changing from “following” and “running in parallel” to “leading”. The scale of the digital consumer market ranks first in the world, and the scale of Chinese netizens ranks first in the world for 13 consecutive years, reaching 1.011 billion in June 2021. The huge scale of netizens promotes the rapid development of a digital green finance industry. The scale and growth rate of the digital economy are among the highest in the world. As an important condition for green development, green finance not only plays the basic role of capital investment and accumulation, but also uses macrocontrol to identify effectively risks caused by environmental factors. Green finance has attracted much attention from all walks of life since it was initiated. However, owing to the late start of green finance in China, many existing problems are still exposed at the exploratory stage. On the one hand, China is making great efforts to transform to green development, and there is a large societal demand, but the supply of green-related capital is difficult to maintain with expanding social demand. On the other hand, green financial investment projects have the problems of a long investment cycle and a low rate of return. In addition, financial audits can optimize the structure of foreign capital and promote the upgrading of China’s structural industries. However, China lacks a dependable green capital trading market, a reasonable regulatory constitution, and a reliable green audit system. At the same time, there are also problems such as information asymmetry and the lack of relevant green finance talents among regions, which lead to “difficulty in landing” green finance projects.

In view of these problems, this paper aims to explore whether a digital economy can effectively improve the investment efficiency of green finance and realize the high-quality development of China’s economy. Through an in-depth study of digital economy and green finance, the OLS model and threshold model are used and reported in this paper, combined with a provincial perspective, to put forward the green development of the digital economy. A literature review on the development and research of green finance from the perspective of digital economy is used to find new ways and new ideas for the development of green finance.

## 2. Literature Review

### 2.1. Definition and Evolution of Digital Economy

As a new economic form, digital economy is increasingly becoming the new kinetic energy of economic growth. By highlighting individual utility, digital economy makes the value of goods return to the subjective value essence. Since the late 1990s, international scholars have been concerned with the development of information technology impact on a social level. Early on, from the perspective of policy research, Rai et al. [2] put forward a measurement method for the social informatization index. Today, the index measurement of digital economy still concerns scholars, and different dimensions are refined to build a comprehensive system. For example, Liu et al. [3], believes that the digital economy has an impact on industrial structure. Based on three dimensions: digital infrastructure, digital industrialization, and industrial digitalization, they construct a comprehensive index system for the development of the digital economy, and measure the development level of China’s digital economy using the time-series-global principal component analysis method. This measurement method reflects the development level of digital economy from the perspective of quantitative indicators. Some scholars are also deeply involved in a one-dimensional digital economy system, with some believing that a digital economy has a positive effect on economic development. The formation of digital information elements deepens the capital production structure by promoting industrial informatization and information industrialization, and extends the capital production structure vertically.

Based on the upsurge of this type of research, the international community pays more attention to the value of a digital economy in the commercial field. Olga et al. [4] believe that a digital economy is also an important determinant of global economic growth and development. Natalia [5] suggests that development of a digital economy contributes to business development and the use of innovative technologies. Mentsiev et al. [6] contributed research indicating that a digital economy enables companies to expand their business scope by using the Internet, thus creating business value. Davies et al. [7] found that a digital economy provides new opportunities for enterprises. These are advantages of a digital economy. In contrast, Lyapuntsova et al. [8] suggest that the risks and challenges of using a digital economy are inevitable, which threaten the development of new models in economic sectors. In this context, digital consumer literacy is very important. Similarly, Li et al. [9] believe that the wide application of digital technology and industry digitalization make it possible for enterprises, consumers, and government to build an entirely new information-based economic system and improve the transaction efficiency of commodity and service activities. This, however, also brings new risks to China’s economic activities and economic environment, which has attracted particular attention in the financial science and technology field of financial industry digitalization.

In view of the current situation of China’s digital economy, Chinese scholars are concerned about the regional imbalance of digital construction among different regions in China. Wang et al. [10] believe that the index system for evaluating the development level of the digital economy should be constructed, and the entropy method should be applied to give weight and descriptive statistics. The Theil index, natural breaks, Moran index, and other methods should be adopted for empirical analysis. The study found that the development level of China’s digital economy increased yearly, but there was significant heterogeneity between the four regions and the five economic zones. Most of the regions were in low-level areas and had a lengthy period of no transition, and therefore maintained a steady state. The problem of insufficient and unbalanced development of the digital economy was still serious. Bai et al. [11] believe that while the digital economy drives economic growth, it also leads to a widening of the income gap through efficiency reform caused by various factors, such as restructuring and upgrading, the industrial intelligence, the digital governance model. The dual macroenvironment of the digital economic development and the decline of the demographic dividend acts on the income distribution effect. There are still some restrictions and problems in digital construction and other aspects. It is necessary to focus on solving the problem of uneven development in the digital economy region, continuously improve the level of digital technology, accelerate the digital collaboration in the industrial chain, and establish and improve the digital economy governance system. From the construction of a digital economic index, Liu et al. [12] suggest that in order to reflect the development level of the digital economy of urban agglomeration, an entropy method can be used to construct the digital economic index of six major urban agglomerations within the digital economic index system. The results show that the differences in digital economy are obvious. The development level of digital economy presents dynamic fluctuations, and the differences in the development level of digital economy among cities within urban agglomeration are obvious. 

Based on these research findings, a digital economy can promote the development of the financial industry. Combined with the research in China, we find that there is regional imbalance in the scale of digital economy among different provinces in China. To explain this phenomenon, this paper selects data from different provinces in China and observes the actual impact of the digital economy, which provides this research with a rational foundation.

### 2.2. Green Finance Concept and Evaluation

As a new financial model, there is no uniform definition of green finance, but through the analysis of existing literature, three main concepts are provided. The first category is environmental finance, which considers the practical problems of environmental protection, pollution control, resource protection, and other green projects by providing financial services (Salazar et al. [13], Cowan et al. [14] and Gray et al. [15]). The second view is that it is a kind of financial innovation, through various financial instruments and products. The third and most recent view is that it is a form of finance, which promotes environmentally friendly investment and cultivates an ecoconscious society through green-oriented credit, securities, insurance and investment, and carbon finance. Under the understanding of these different concepts, research on green finance is also deepening with a focus on qualitative analysis and conceptual understanding of green finance. For example, it is believed that the green credit policy is helpful for improving the allocation efficiency of financial resources, and guides funds from industries with high energy consumption and high pollution to environmental protection industries, thus improving environmental quality. It also shows that green financial capital and environmental governance investment play an important role in alleviating the derivative problems of environmental deterioration. Through the allocation of credit resources and compared with previous qualitative analysis, research related to quantitative analysis of the development level of green finance is relatively lacking. Existing research mainly uses a single indicator to evaluate, such as the progress of low-carbon financial flows and the amount of green credit and green investment to represent the development of green finance.

Although these single conceptual indicators can reflect the development of green finance to a certain extent, they have difficulty revealing the entire view of green finance. In order to solve the limitations of the existing indicators, some scholars further build an evaluation system to measure the comprehensive level of green financial development. Referring to the composite system of green finance development, an index system has been put forward to measure the development of China’s green finance from five aspects: green credit, securities, insurance, investment, and carbon finance. After establishing the index system of green finance, people give more attention to the connection between green finance and regional high-quality development. At the same time, international scholars also put forward that the development of green finance needs to rely on the help of society and enterprises to establish the awareness of green finance. Salzmann [16] believes that, also relying on the financial market, the framework of green finance emphasizes that the main connection between the capital supply and demand side, the financial market, and financial intermediaries lies in social responsibility and the green management concept. Labatt et al. [17] believes that green finance is a kind of financial instrument to improve environmental quality and reduce environmental risks, and is based on the market. Chinese scholars have also proposed to establish China’s green financial system to help the social economy achieve green transformation. For example, Ma [18] believes that the establishment of a green financial system in China will help to start new growth points; enhance the economic growth potential; accelerate the green transformation of industrial structure, energy structure, and transportation structure; enhance the technological content of the economy; and ease the financial pressure from environmental problems. Liu et al. [19] believe that green credit is an important financial policy to achieve a high-quality economy and will use more market-oriented economic measures to control the environment. Yuan [20] believes that financial support is helpful to the development of green finance from the aspects of capital support, enterprise transformation efficiency, resource allocation, and technological innovation. Chinese scholars believe that green finance is inseparable from financial support and needs help from different levels. Zheng et al. [21] believe that the establishment of an external institutional environment for the development of regional green finance can improve the visibility of enterprises’ transformation risks; promote green finance in layers based on the degree of environmental externalities; and strengthen the identification, analysis, and management of environmental risks. On building a green financial system, Chinese scholars have found that while building China’s financial system, it is also necessary to join the market supervision and strengthen the ability to identify risks and crises.

Scholars both in China and abroad have limited their research to the macrolevel. At the same time, the criteria for building a green financial indicator system have not been unified. Scholars in different fields focus on indicators differently. Through literature research and combining with the theme of the article, this paper focuses on green financial indicators from four aspects: green credit, green investment, green insurance, and government support.

### 2.3. Research on the Impact of Digital Economy on the Efficiency of Green Finance

It is of great significance for initial scholars to study how environmental regulation affects green technological innovation and the relationship among environmental regulation, digital economy, and green finance, which can encourage enterprises to carry out technological innovation and realize green economic development. For example, Grossman et al. [22] found that the relationship between some environmental pollutant emissions and economic growth is similar to the Kuznets curve, showing an inverted “U” curve. Chinese scholars are also concerned that the economy needs green development, and they use scientific and technological means to promote efficient economic development. He [23] suggests that the improvement of China’s economic development quality mainly comes from the improvement of green development, economic efficiency, and social welfare. Wang et al. [24] believe that from the perspective of China, environmental regulation can significantly promote the quality of economic growth, but technological innovation is needed to promote high-quality economic development. Therefore, under the new development concept, only by following the guidance of environmental regulations and adopting digital economy technology to achieve green upgrading can enterprises or industries resolve the double pressure from the government and the market.

The continuous development of a digital economy has attracted people’s attention. Both digital economy and green finance are related to high-quality economic development, and there is little research on linking them together. A digital economy can enable consumers to distribute their own renewable energy and effectively use limited resources to improve the efficiency of a green economy. Ciocoiu et al. [25] suggest that the integration between digital and green economies can lead to a new model, and creates some opportunities for sustainable development and economic recovery. Zheng et al. [26] believe that as two very important economic forms in the future, there is a close relationship between digital economy and green finance. Xue [27] believes that the digital economy represented by the Internet of Things can promote the development of a green economy and provides a new growth point for the sound development of China’s economy. Qian [28] pointed out that a green economy and a digital economy have mutual needs and require mutual assistance. Only by fully cooperating with green development and digital development can the potential of China’s economy be released to the greatest extent. Therefore, scholars are concerned that the digital economy will have an impact on the development of green finance, and believe that they have great development potential in the future.

Many scholars have given attention to the fact that the development of a digital economy can make the economy go green, but they have not tested quantitatively whether digital economy can have an impact on the development of green finance and what is the clear relationship between digital economy and our economic investment. Moreover, there are very few articles that directly study the relationship between digital economy and green finance, which is also the research value of this paper, to discuss whether the development and application of a digital economy can really and effectively improve the investment and development efficiency of green finance.

## 3. Research Methods and Research Assumptions

### 3.1. Theoretical Analysis and Research Hypothesis

There is little research on the relationship between digital economy and green finance. After studying the relevant literature, our data type is set as the panel data of 30 provinces in China. Therefore, choosing the static OLS model for regression analysis can satisfactorily reflect and describe whether the relationship between digital economy and green finance is significant, and therefore meet the basic assumptions of this paper. At the same time, we found that the previous empirical analysis of the digital economy’s influence factors on green finance is based on a hidden premise that there is no difference in the factor endowments of all regions in China. In fact, the digital economy has to be restricted and influenced by many factors if it is to give full play to its innovative features. Therefore, by using the threshold model—considering the influence of external environmental factors, adding threshold variables, and again fitting the relationship between digital economy and green finance—the results of data generation are more scientific and specific problems can be found. Two external environments are selected: regional industry scale and green financial audit. Because the larger the industry scale, the better the digital facilities of green finance are generally built. The choice of green financial audit is because, while the scale is expanding, the traditional audit mode cannot meet the development of green finance, which lags behind and is passive. Therefore, only by joining the external condition of a green financial audit, standardizing the operation behavior of market participants as much as possible, strengthening the legal construction of financial industry, and supervising the effectiveness of policy implementation, can we prevent and control the systemic risks of green finance to the maximum extent. These two methods are in line with the research assumption of this paper; thus, this paper analyzes the relationship between them and the two threshold variables. On the one hand, the digital economy plays an increasingly important role in China’s financial industry and economic growth, showing strong vitality and becoming one of the important forces to promote China’s economic development. On the other hand, green finance is an important driving force to realize “green development”, which requires the response and support of green finance [29].


*The mechanism of the influence of digital economy on the promotion of China’s provincial green finance is complex; thus, can the rapid development of China’s digital economy effectively improve the investment efficiency of China’s provincial green finance? Based on our research methods and the answers to these questions, we make the following assumptions:*


**Hypothesis** **1.**
*The digital economy improves the efficiency of green financial resources in China’s provinces under the premise that other influencing factors are controlled.*


Regional Industry Scale

The concentrated expression of regional industry scale is that the larger the regional industry scale, the more motivated it will be to carry out digital infrastructure construction, which is the basic link for vigorously developing the digital economy. The research shows that the better the digital infrastructure, the more it can promote the development of green financial industry structure, and the better the digital economy of cities with large-scale industries will develop [30]. This paper assumes that the development of digital economy depends on the scale of regional industries. Only when the regional industrial scale reaches a certain level can the digital infrastructure construction of green finance be rapidly promoted [31]. The purpose of green finance is to penetrate the development concept of environmental protection into the whole environmental protection industry, so as to realize the optimization and upgrading of industrial structure and the sustainable development of economy [32]. On the contrary, if the industry size of the region is below a certain threshold, the region may use its limited resources to invest in other matters, which is not conducive to the construction of digital infrastructure and the development of the green features of the digital economy. The scale of regional industries plays a “screening” function in the process of digital development. The smaller the scale of regional industries, the less favorable it is for the investment in digital infrastructure construction, and it has a negative effect on the promotion of green finance by the digital economy. However, the cities with larger scale industry in the region have obvious advantages in terms of the development degree of digital infrastructure, capital sources, and digital talents concentration, and can better display the green features of the digital economy. Based on this analysis, the following assumption is made.

**Hypothesis** **2.**
*Under the premise that other influencing factors are controlled, there is a threshold effect of regional industry size on the impact of digital economy on green financial efficiency.*


2.Green Financial Audit Support

A green financial audit can exploit an innovative financial audit mode, strengthen the supervision before and during “financial greening”, and curb or avoid the green risks of financial activities from the source [33]. If the green economy wants to stimulate greater growth potential and needs strong financial support, the role of the financial audit cannot be ignored. A green finance audit can not only supervise the relevant measures of the green finance industry, but also promote the development of regional green finance. Establishing and improving a standardized financial system, encouraging enterprises to make technological innovation to transform into green industries such as high-tech environmental protection, and guiding consumers to establish green concepts such as environmental protection and green consumption, all produce capital flows to green environmental protection undertakings. The greater the government’s financial investment in environmental protection, the more inclined the policy is to green development—the more opportunities there are. When a region has more financial audit support, its financial environment improves. Moreover, it has the advantage of gathering relevant professionals and can better display the characteristics of green economy. Based on this analysis, the following assumption is made.

**Hypothesis** **3.**
*Under the premise that other influencing factors are controlled, there is a threshold effect of green financial audit support for the impact of digital economy on green financial efficiency in China’s provinces.*


### 3.2. Model Construction

#### 3.2.1. OLS Model

At present, the digital economic indicators of institutional statistics are mainly at the national level, and a lack of long-term measurement data are at the provincial level. Therefore, according to the research ideas and research assumptions, digital economic variables are selected as the main explanatory variables, and various influencing factors that can affect the efficiency of the green economy are selected as the control variables to establish the multiple linear regression model, as shown in Formula (1).
(1)$$GF_{it}=a_{0}+a_{1}Number+\beta X_{it}+\mu_{it}$$
where i is the province, t is the time year, and GF represents green finance. Number represents the digital economic index, $a_{0}$ is the intercept term, $a_{1}$ represents the regression coefficient of explanatory variable, $X_{it}$ is the control variable, β is the regression coefficient of each control variable, and $\mu_{it}$ the is error term.

#### 3.2.2. Threshold Model

Hypothesis 2 and 3 are proposed in the research hypothesis section above. This paper sets a panel threshold model to verify the relevant assumptions. This paper uses Hansen’s (1999) method for reference, sets the regional industry size and financial support as threshold variables, and thoroughly analyzes the nonlinear impact of digital economy on green finance in domestic provinces. The panel threshold model is set in (2) and (3), as follows:(2)$$\begin{array}{l} GF_{it}=a_{1}Number_{it}\times I(HYGM_{it}\gamma_{1})+a_{2}Number_{it}\times I(\gamma_{1}<HYGM_{it}\gamma_{2})+\cdots +a_{n}Number_{it}\times \\ I(\gamma_{n-1}<HYGM_{it}\leq \gamma_{n})+a_{n+1}Number_{it}\times I(HYGM_{it}>\gamma_{n})+\theta Z_{it}+\mu_{i}+\varepsilon_{it} \end{array}$$
(3)$$\begin{array}{l} GF_{it}=a_{1}Number_{it}\times I(GFA_{it}\gamma_{1})+a_{2}Number_{it}\times I(\gamma_{1}<GFA_{it}\gamma_{2})+\cdots +a_{n}Number_{it}\times \\ I(\gamma_{n-1}<GFA_{it}\leq \gamma_{n})+a_{n+1}Number_{it}\times I(GFA_{it}>\gamma_{n})+\theta Z_{it}+\mu_{i}+\varepsilon_{it} \end{array}$$

In the threshold model, the threshold variable is set as regional industry size ($HYGM_{it}$) and green financial audit support ($GFA_{it}$), respectively, according to assumptions 2 and 3; γ is the corresponding threshold value, $\mu_{i}$ is the individual fixed effect; I(•) is an index function; and $\varepsilon_{it}$ is a random error term. If true, the value in parentheses is 1; otherwise, it is 0.

### 3.3. Core Explanatory Variables

#### 3.3.1. Digital Economy

At present, the major domestic institutions that publish digital economy are: Tencent Research Institute Digital China Index, Alibaba and KPMG Digital Economic Development Index, Caixin Think Tank, and China Information and Communication Research Institute Digital Economic Index. The focus of what each agency publishes is different. Similar to digital China launched by the Tencent Research Institute, it is mainly divided into five indexes: digital China, digital industry, digital government, digital culture, and digital life. Tencent’s digital data are shown in Table 1, which are mainly based on the data of its servers and its industries. The research level is not sufficiently rich, but its huge user base provides an accurate understanding of the development trend of the digital economy.

Ali Institute and KPMG jointly released the “2018 Global Digital Economic Development Index” report. The global digital economy development index covers 150 countries and regions, and depicts the level, structure, and development path of the digital economy through five dimensions: digital infrastructure, digital consumers, digital industry ecology, digital public services, and digital scientific research. The specific index system is shown in Table 2.

This paper selects the digital economy index data of 30 provinces in China (except Tibet, Hong Kong, Macao, and Taiwan) from 2012 to 2018, refers to the index data constructed by Caixin Think Tank and Tencent Digital Center, and obtains the digital economy index. Because there are too many digital economy data missing in Tibet, this paper does not include it as a research object. Some data of other provinces after 2018 are missing and have not yet been made public. Therefore, this paper selects the provincial data up to 2018, and a small amount of missing data is filled with hot cards.

#### 3.3.2. Green Finance

Technological innovation in the era of digital economy is conducive to the development of green finance. At present, the index construction of green finance is mainly embodied in four dimensions, namely, green credit, green investment, green insurance, and government support. Four specific secondary indicators are selected to describe the development of green finance in China’s provinces in the era of digital economy. In this paper, the six-year data of Green Finance from 2012 to 2018 are selected from the China Statistical Yearbook, statistical yearbooks of various provinces, and the China Insurance Yearbook, and are calculated by using the entropy method. It should be noted that due to the lack of data for some provinces, the lack of data for some years is replaced by the average value of nearly five years; for example, the green credit lacks the data for 2017 and the green insurance lacks the data for 2001. The specific indicators are shown in Table 3.

#### 3.3.3. Control Variables

The control variables involved in the empirical design of the research presented in this paper are the level of economic development; the per capita GDP (PG) of each province; the industrial structure (Structure), mainly expressed by the proportion of the secondary industry in GDP; the urbanization (Urban) expressed by the proportion of urban population in the total population; and the foreign trade (Trade) measured by the level of foreign trade in each province by the total import and export of each province.

#### 3.3.4. Threshold Variables

Regional Industry Scale

This paper uses the local financial expenditure on science and technology from 2012 to 2018 to express the scale of digital economy industry in each region, because large industrial enterprises have advantages in research and development of digital economy technology innovation. Therefore, the larger the industry scale in the region, the more motivated the region will be to invest in the construction of a digital economy. Therefore, the green financial efficiency may be affected when the industry size of the region exceeds the threshold.

2.Green Financial Audit Support

This research selects the data on local financial and environmental protection expenditures of each province from 2012 to 2018. Financial input and audit support for environmental protection expenditures also reflect the support of government policies for green development. The government is the key to solving the dilemma between ecology and growth, and realizing green development. The 19th National Congress of the Communist Party of China has made an in-depth interpretation and long-term planning of green transformation, green economy, and green finance from the perspective of strategic vision. Under the call for government policies, China is forming a strong green financial development trend to guide and encourage social capital to enter the green industry. Therefore, the government financial audit support has played a significant role. This research selects the variable of local financial environmental protection expenditure. When a region invests more in environmental resources protection, the space for its green financial development also increases.

### 3.4. Empirical Test and Analysis

#### 3.4.1. Basic Regression Analysis Test

The research described in this paper uses stata16.0 software to conduct empirical tests. Stata is a statistical analysis system (English: statistical analysis system) of a company in the United States. It was written and formulated by two graduate students of Biostatistics at North Carolina State University. Firstly, it tests the significance between variables and then uses the OLS method to conduct regression analysis on static panel data. Through literature analysis, this research uses the article to gradually add control variables to conduct double fixed effect regression analysis. The specific inspection results are shown in Table 4.

From the regression analysis results of models (1)–(7) in Table 4, it can be seen that with the continuous addition of control variables, the goodness of fit of the model continues to improve, which indicates that the selection of core explanatory variables and control variables is scientific to a certain extent.

#### 3.4.2. Robustness Test

To test the robustness of the regression results, the sample is adjusted and the same empirical method is again used for regression analysis. The specific approach follows: Firstly, the sample year is processed, the sample data of 2012, 2015, and 2018 are removed, and the empirical analysis is again performed. Secondly, as the core explanatory variable digital economy is processed, the maximum value and the minimum value are removed before re-estimation. Finally, to overcome the nonrandom interference to the model estimation, the green financial maxima and minima of the explained variables are still removed by removing the extremum and then regressed. All of the above regression results show that they are basically consistent with the previous empirical conclusions, and the regression results have good robustness characteristics.

#### 3.4.3. Threshold Effect of Digital Economy on Green Finance

Threshold Effect Test

The foregoing empirical analysis of the digital economy’s impact on green finance is based on the hidden premise that all regional factor endowments in China are not different. In fact, the digital economy needs to be restricted and influenced by various factors to develop its innovative features. At the same time, the digital economy and green finance can exhibit nonlinear characteristics. In order to respond to research Hypotheses 2 and 3 proposed above, this research analyzes the nonlinear effects of digital economy on China’s total industrial green factor productivity. Based on the relevant literature, this research uses regional industry scale and green financial audit support as threshold variables to analyze the threshold effect. The specific inspection results are shown in Table 5.

Before the threshold test, it is necessary to verify whether the model has a threshold and its significance level. In this work, the bootstrap method is used to simulate the threshold value of 300 searches, and the results are shown in Table 6. It can be seen from Table 6 that the threshold variables have a threshold effect in the relationship between green finance and digital economy, in which there is a double threshold for regional industry size (STP) and a single threshold for financial audit support (GFA).

2.Threshold Regression Study

In this research, the threshold test is carried out for the two models of Hypotheses 2 and 3 in the design. The specific parameter estimation results are shown in Table 7.

Table 7 shows the results of threshold regression estimation when the regional industry size and institutional environment are the threshold variables. As can be seen from Table 7, when the industrial structure of the regional industry scale does not cross the first threshold, the impact of digital economy on green financial stocks is not significant. With the improvement of industrial structure, the promotion effect between the two gradually appears. When the industrial structure crosses the second threshold, the impact of digital economy on green finance is still significant at 1%. This shows that with the progress of infrastructure in the digital economy, green finance is more willing to “green” and reduce emissions. As for technological innovation, it can be observed from Table 7 that with the improvement of technological innovation, the coefficient between the two becomes significant at 5% and 1%, indicating that the improvement of regional industry scale plays an important mediating role in the relationship between green finance and digital economy, which also emphasizes once again that the development of digital economy is indeed beneficial to the implementation of green finance policy.

Examining the green financial audit support again, only the first threshold is crossed, and the digital economy plays a positive role in driving green finance. When the second threshold is not crossed, the effect between the two is weakened, and the digital economy has a significant inhibitory effect on green finance. This reflects that the current economic development in our country is accompanied by the government’s neglect of the implementation of environmental protection projects in pursuit of political achievements and enterprises’ pursuit of profit maximization, thus inhibiting the green development of enterprises. It can be seen that the current high level of economic development in our country is not conducive to the implementation of green finance in the digital economy. There are still problems such as insufficient risk prevention, control capability, and incomplete green audit content.

## 4. Discussion

In-depth study of digital economy and green finance related literature can master the knowledge of related fields, and can also play a reference role for our own research. As the digital economy can promote a large number of scientific and technological innovations, the development of digital economy is of great significance to the transformation and upgrading of green finance, which can effectively promote the high-quality development of the economy and contribute to the transformation and development of the traditional financial industry. In the part of literature, we focus on green finance, digital economy, and the relationship between green finance and digital economy, and discuss the influence of digital economy on the development of green finance, and put forward the views of this paper. This paper finds the shortcomings of the research, uses digital economy to promote the high-quality development of China’s provincial green finance, and at the same time provides reference for the research in related fields.

In recent years, scholars have paid attention to the digital economy’s contribution to the high-quality green development of the economy, which is directly conducive to enhancing the vitality of urban green development [34,35]. The typical characteristics of digital economy, such as permeability, platformization and sharing, can effectively empower traditional industries, improve resource utilization rate and raise the level of digitalization of finance [36,37]. We conducted more detailed research on the economic field, focusing our attention on the traditional financial industry. At present, the research on green finance in digital economy only mentions that the application of digital technology is helpful to the development of green finance [38], and lacks targeted attention in this field. Few authors directly discuss the impact of digital economy on green finance. Part of the research examines the impact of financial digitalization on the development of green finance from the micro-enterprise financing level [39], while this paper discusses whether digital economy can improve the investment efficiency of China’s provincial green finance from the macro-angle, enriching the research in related fields. With the help of digital economy technology, realize the high-quality development of green finance and promote the development of China’s green finance industry.

There are some limitations in this study. Future research will make up for the shortcomings of this study and provide reference direction for future research. The first limitation is that the data sample only reports the data of 30 provinces in China, and because some data are not public, the data research can be further improved to improve the accuracy of the conclusion. In addition, as a developing country, China is different from other developed countries and other regions in the impact of digital economy on green finance, including Europe, North America and Oceania. Therefore, our future research needs to improve its universality in the world. Another limitation is that the research object of this paper is the data of 30 different provinces in China. It is necessary to establish a unified index system, which cannot be completely accurate to digital economy and green finance. In future research, with the publication of data and the development of the industry, the accuracy of the index system will be continuously improved. There are still differences between the international index system and the Chinese index system. The Chinese index system selected in this paper may not be suitable for international universality. In the next research, we should also establish an index system that conforms to the global norms.

The research results show that the application of digital technology in green financial business can not only help regulators improve the efficiency of investment and audit in green financial business, thus further promoting the development of China’s green financial market, but also help financial institutions to optimize green financial business, reduce the management cost of green financial business, and thus enhance the development power of green financial business of financial institutions. The deficiency of the influence of digital economy on the investment of green finance development, and the corresponding policy suggestions are given, which provide a reference for scholars in China and other regions.

## 5. Conclusions

In this research, OLS and threshold models are used for data analysis, and the impact of digital economy on China’s provincial green finance is thoroughly studied. The main conclusions follow: First, a digital economy can significantly improve the efficiency of China’s provincial green finance. A digital economy can not only hasten the transformation of China’s traditional financial industry to a green financial industry, but it also has the function of low carbon and environmental protection, which can become the new kinetic energy of China’s economic growth under the background of the green economy era [40]. Secondly, the digital economy has a double threshold effect on the provincial green finance, with the regional industry scale as the threshold variable. Therefore, the larger the industry scale in China, the more motivated it is to build digital infrastructure, so as to better obtain the green innovation advantages brought by the digital economy. Thirdly, the digital economy has a single threshold effect on China’s green finance, which is supported by green financial audit. Although the digital economy contributes to the growth of provincial green finance, China’s investment in green audit supervision is insufficient, and the regression conclusion is basically consistent with the previous empirical results. Through the OLS model test, from the regression analysis results of models (1)~(5) shown in Table 4, it can be seen that with the gradual addition of control variables, the goodness of fit of the model is constantly improved, which indicates that the selection of core explanatory variables and control variables is scientific. According to the results of the regression analysis, it can be found that among the core explanatory variables, there is a significant positive correlation between the digital economy and the green financial index, mainly because in the static environment and the increase in China’s economic development level, all kinds of fixed assets investment and foreign trade level has strengthened the continuous expansion of the investment scale of the green financial industry. On the whole, the development of the digital economy plays a significant positive role in promoting the efficiency of China’s green finance, which is reflected in all regions of the country. This shows that China will upgrade the digital transformation of green finance in the process of building the digital economy era, which will continuously improve the investment efficiency of green finance. At the same time, we found that although the digital economy has a significant role in promoting green finance, there are economic differences among provinces in China, insufficient investment in digital economy, and imperfect infrastructure construction of digital economy.

A further threshold test can reflect well the specific problems in the process of digitalization of green financial industry. The impact of digital economy on China’s green financial investment has double threshold effects of the following regional industry scale threshold variables. Therefore, the larger the industry scale in China, the more motivated it is to build digital infrastructure, so as to better obtain the green innovation advantages brought by digital economy. This shows that with the progress of digital economy in industry scale, the willingness of green finance to “green” adjustment and environmental protection and emission reduction in financial industry is stronger. Therefore, the impact on green finance will be increasingly stronger. However, the abuse of digitalization also emerges in an endless stream. We should not only increase the scale of digitalization in the green financial industry, but also give attention to improving the efficiency of digital application in the green financial industry. In addition, the impact of digital economy on China’s green financial investment has a single threshold effect with green financial audit support as the threshold variable. When the support of green financial auditing has not crossed the second threshold, the role between them is weakened, and the digital economy has inhibited green finance. This reflects that while China’s current economic development is accompanied by the neglect of the implementation and supervision of environmental protection projects by some local governments in pursuit of political achievements and the pursuit of profit maximization by the financial industry, the funds invested are insufficient, thus inhibiting the development of green financial industry. It can be seen that China’s current level of financial auditing related to green aspects is not conducive to the implementation of green finance by digital economy. Moreover, there are still some problems in China, such as the lack of risk prevention and control ability of the green financial industry and the nonstandard green audit system, which lead to the green financial industry not attracting more investment.

Overall, the empirical test results show that the development of a digital economy has generally improved the investment efficiency of China’s green finance, and it can be confirmed in the three research hypotheses that digital economy plays a significant positive role in improving the efficiency of green finance.

## 6. Policy Suggestions

With the penetration of digital economy in economic society, the global economy is transforming from service-oriented to digital-driven. Digital technology has also brought about new trade patterns and trade targets change in the trade field. Accelerating the construction of digital platform infrastructure can provide multi-dimensional data support for financial institutions in monitoring, early warning, assessment and disposal of environmental, climate and social risks [41]. Developing green finance, slowing down and adapting to climate change, and promoting comprehensive green transformation of economic and social development not only reflect China’s firm determination to develop with high quality, but also demonstrate China’s rapid development in promoting the construction of Community of Shared Future for Mankind’s image as a big country with responsibility and responsibility [42], which is exactly the catalyst for the rise of green finance in China. Through the construction of a digital economy, the expansion of the green financial system is an important way to transform China’s economic development. The traditional economy and the emerging digital economy, which are promoted by science and technology, are greener and lower carbon [43]. According to the research results of this paper, the following suggestions are made.

### 6.1. Accelerate the Construction of Digital Infrastructure, and Narrow the Development Imbalance of Green Finance Regions

It is necessary to fully tap the green value of the digital economy, accelerate the rapid development of 5G projects, optimize the business environment in various regions, and promote the rapid growth of green finance in the provinces [44]. The empirical analysis of the influence factors of the digital economy on green finance shows that the digital economy is restricted and influenced by many factors if it wants to play its innovative features, so there is an imbalance in regional factors among different provinces in China. We will focus on building a solid industrial foundation for the digital economy, step up research on key technologies, and raise the level of development of the digital economy’s basic industries and industrial modernization. It is the key to reduce the imbalance in the development of green finance. Therefore, economic development and environmental protection are not contradictory but mutually reinforcing. The digital economy has such characteristics that it cannot only bring into play the green value of the digital economy, but also enable the digital economy to empower the traditional industrial enterprises. As the main engine of China’s economic growth, the digital economy plays an important strategic role in building a new development pattern. Continuously releasing new market potential and becoming an important leader in independent innovation, trade innovation, employment creation, and an important provider of public services are crucial breakthroughs in building a new development pattern. Studying the relationship between digital economy and green finance and creating a new format of green digital industry can better help our country to achieve the goal of sustainable development. It is also an important starting point for the financial industry to help supply-side structural reform and better serve the construction of ecological civilization [45]. Additionally, at the same time, the digital economy helps China achieve the goal of sustainable development, and provides reference suggestions for the regional imbalance of green finance in other countries worldwide.

### 6.2. Improve the Utilization Efficiency of Digital Economy and Form a New Format of Green Financial Industry

Green finance has become an emerging growth point of financial institutions’ business. The application of digital economy technology in development has enabled the integration and innovation of green finance and traditional finance. In the previous research results, it is found that there is a double threshold phenomenon in regional industry scale, and the application efficiency of digital economy in green finance industry is not high enough, which leads to the industry scale gap between different regions. A standardized green financial system can realize the development of the industry and ensure the high quality of green finance. Therefore, green finance, as a modern economy, plays a core guiding role in optimizing the rational allocation of resources. Green finance can effectively help prevent and control environmental pollution and promote green economic transformation. Green finance can not only provide funds for green projects, but also alleviate the risk of maturity mismatch of green projects by issuing green bonds and asset securitization, and effectively restrain the production of pollution projects by restricting loans and stopping loans. Therefore, while actively playing the leading role of the market, the government should also give attention to the role of audit supervision, create a fair and just green financial environment, and add to the development and construction of a green financial system. These actions will achieve the three major objectives: reduce green investment costs, increase the investment costs of polluting projects, and raise the awareness of social responsibility of enterprises and consumers, which together will cultivate the relevant basic conditions for the rapid and innovative development of green finance.

### 6.3. Improve the Development and Construction of the Green Financial System and Strengthen the Role of Audit Supervision

The digital economy can achieve high-quality development in the post-epidemic era in China and optimize the cross-regional allocation of digital resources. Therefore, the world should establish a standardized green financial system, realize industry transformation and upgrading, and ensure the high-quality development of green finance. Therefore, as a modern economy, green finance plays a core guiding role in optimizing the rational allocation of resources, and the implementation of green financial policies is inseparable from the support of the digital economy. Only by continuously promoting the institutional innovation related to digital economy can China’s digital economy be expanded and its developmental quality improved. To this end, as soon as possible, China should introduce and perfect the Personal Information Protection Law and the Data Safety Law, together with other laws and regulations, to explore the protection of intellectual property rights in the digital economy and the security of personal privacy data. Finally, it is necessary to strengthen the development of a digital talents team. At present, talents related to the domestic digital economy and green finance team are quite scarce, and there is a phenomenon that “one understands economy but does not understand digital technology, and one understands digital technology but does not understand economy”. At present, there is an urgent need for domestic multidisciplinary professionals who understand both economy and digital technology. Universities and research institutes should establish digital economy-related and green finance-related majors as soon as possible to improve the effectiveness of related personnel training.

## Figures and Tables

**Table 1 ijerph-19-08884-t001:** Tencent digital data indicators.

Index Name	Index Source
Digital China	Digital Development Level of Provinces and Cities in China
Digital industry	It covers the WeChat public platform, WeChat payment, QQ payment, Tencent Cloud, Tencent map, enterprise WeChat data, and other Internet enterprise data such as JD.COM, Didi, Meituan Dianping, Ctrip, and Pinduoduo. It measures the activity of the digital industry and outlines the development trend of industrial Internet in various regions.
Digital culture	The digital culture index collects data from Tencent Games, Tencent News Client, Tencent Video, Tencent Animation, Yuewen Group, Tencent Music Group (TME), Snack Video, and Cat’s Eye movies to measure the vitality of the regional digital culture market and the digital culture consumption potential.
Digital government	The digital government index measures the development level of regional government mobility and the activity of online government services based on the government-related WeChat official account and WeChat city-service data.
Digital life	The digital life index collects data from QQ, WeChat, and Tenpay to measure the social activity in the region and the digitalization level of offline life services.

**Table 2 ijerph-19-08884-t002:** Five dimensions of digital economy published by Alibaba and KPMG.

Index Name	Subindicators
Digital infrastructure	Data, quality, and price of the facility. Penetration rate of network and mobile Internet, network speed, penetration rate of mobile terminals and consumption capacity of mobile terminals.
Digital research	Number and quality of ICT-related research, number of ICT-related patents, and index of highly cited papers
Digital public service	Utilization level of government digital technology and the popularity rate of e-government services
Digital consumers	Penetration of digital technology into consumers, social networks, online shopping, and mobile payment penetration rate
Digital industry ecology	Degree of digitalization of the industry, the penetration rate of digital technology into the industry, the level of application of new technologies by enterprises, and the proportion of digitalized industries

**Table 3 ijerph-19-08884-t003:** Level 1 indicators of the green financial indicator system.

Level 1 Indicators	Indicator Name	Indicator Description	Indicator Attribute
Green credit	High energy consumption industry interest expense proportion	Six high energy consumption industrial interest expenses\total industrial interest expenses	-
Green investment	Investment in environmental pollution control as a proportion of GDP	Investment in environmental pollution control\GDP	+
Green insurance	Depth of agricultural insurance	Agricultural insurance income\total agricultural output value	+
Government support	Proportion of expenditure on financial and environmental protection	Expenditure on financial environmental protection\general budget expenditure of finance	-

**Table 4 ijerph-19-08884-t004:** Static panel test results (OLS).

Variable	Model (1)	Model (2)	Model (3)	Model (4)	Model (5)
Number	−0.008 ***	−0.005 ***	−0.005 ***	−0.005 ***	−0.006 ***
	(−4.30)	(−6.71)	(−4.31)	(−4.51)	(−4.78)
PG		0.000 **	0.000 ***	0.000 **	0.000 ***
		(2.72)	(2.77)	(2.62)	(2.92)
Structure			−0.072	−0.057	−0.071
			(−1.02)	(−0.76)	(−1.00)
Urban				−0.268	−0.211
				(−1.16)	(−1.08)
Trade					−0.000 *
					(−1.92)
Province	YES	YES	YES	YES	YES
Year	YES	YES	YES	YES	YES
Constant	0.703 ***	0.400 ***	0.388 ***	0.597 ***	0.612 ***
	(5.70)	(6.28)	(5.85)	(3.58)	(4.01)
N	210	210	210	210	210
R^2^	0.560	0.756	0.757	0.762	0.812

Note: ***, **, * are significant at confidence levels of 1%, 5%, and 10%, respectively, with standard errors in brackets (same as the following table).

**Table 5 ijerph-19-08884-t005:** Threshold effect test.

Threshold Variable	Threshold Number	*p*-Value	Variance Ratio	BS Number	Critical Value		
					1%	5%	10%
Regional industry scale	Single threshold	0.0033	76.94	300	56.4119	41.4159	32.5885
	Double threshold	0.0000	116.27	300	35.0507	25.1987	20.4790
	Triple threshold	0.1700	40.11	300	160.1672	132.1154	107.4622
Green financial audit support	Single threshold	0.0133	36.49	300	37.0922	23.7211	21.1802
	Double threshold	0.5133	7.90	300	69.0719	39.5777	28.5482
	Triple threshold	0.9133	3.30	300	29.8156	20.2369	16.2760

**Table 6 ijerph-19-08884-t006:** Threshold estimates.

Threshold Variable	Threshold Number	Estimated Value	Confidence Interval
Regional industry scale	Double threshold	34.9000	[26.8000, 37.1000]
		47.3000	[44.3000, 52.5000]
Green financial audit support	Single threshold	358.7000	[337.8900, 363.3800]

**Table 7 ijerph-19-08884-t007:** Threshold model estimation results.

Threshold Variable	Model 6	Model 7
	Regional Industry Scale	Green Financial Audit Support
PG	2.31 × 10^−6^ *** (18.07)	2.70 × 10^−6^ *** (17.46)
Structure	−0.0836268 * (−1.96)	−0.1190686 ** (−2.38)
Urban	0.0147933 (0.29)	−0.772259 (−1.35)
Trade	−2.28 × 10^−10^ *** (−5.03)	−3.49 × 10^−10^ *** (−7.57)
Number (STP ≤ 34.9000)	−0.0047143 ** (−2.22)	
Number (34.900 < STP ≤ 47.3000)	0.2925535 *** (4.04)	
Number (STP > 47.3000)	0.539655 *** (8.46)	
Number (STP ≤ 358.7000)		−0.0046724 * (−1.90)
Number (STP > 358.7000)		−0.0220084 * (−1.89)
Constant term	0.4377224 *** (3.11)	0.5042788 *** (3.10)

Note: *, **, *** represent the significance of 10%, 5%, and 1%, respectively, with t statistics in brackets.

## Data Availability

Not applicable.

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
