# Peer review of "The Impact of Digital Economy on the Efficiency of Green Financial Investment in China’s Provinces"

_ijerph, 2022, doi:10.3390/ijerph19148884_

Round 1

Reviewer 1 Report

Dear authors,

I would say that this is an interesting study. However, I believe that there are some rooms for improvement. Please find the following points for further refinement and/or adjustments:

1) Some unclear/misleading sentences - consider revising (e.g., the abstract tests the impact efficiency of digital economy on green financial investment in China by static panel OLS and threshold model method...) and many others.

2) Instances of unsubstantiated claims. "The digital economy can significantly improve China's green financial efficiency as a whole..." It could have been better if further elaborations are made in the relevant sections and examples are provided (not necessarily in the abstract itself). But drawing from what is being claimed in the abstract would enhance readability. I noticed that the authors did make some references to, e.g., improve the efficiency of market operation, etc. But they were all mentioned in passing without making further details within the context of China which I believe that there are a lot of interesting things to tell about. 

3) The literature review could have been better. Rather than merely presenting definitions, concepts etc. that are quite straightforward and rather simplistic for this kind of study, the authors may want to consider making some analysis of the previous study of similar nature to critically describe the issues at hand. What is it the main issues that are happening in other contexts and why are they happening or not happening in China. How is China different or similar, etc. Please highlight the gaps that would further justify the need for further research.

4) How many provinces were really analysed? All? The closest information that could be deciphered was "...due to the lack 317 of data in some provinces, the lack of data in some years is replaced by the average value 318 of nearly five years". Kindly specify which provinces were analyzed and why (why not) they were (not) analyzed.

5) Please justify the use of the OLS model and threshold model. Although the authors did mention that they are used to deeply study the impact of digital economy on provincial green finance, but little is know why not other models. I believe there is a need to provide some strong justifications to this so that readers might understand the plausibility of engaging such models over the others.

6) Quite a number of instances that the facts and claims were made without any specific citations and references. Indeed, some citations and references are not properly cited and referenced. E.g., L.P.Rai (2000), Bai Peiwen and Zhang Yun (2021), and Yuan Huaxi and others (2019). Only surnames are required. Please improve on these by using the approved referencing convention of this journal.

7) Many recommendations posed in the final part of the paper were not well-thought-of. They were just mentioned in passing without much discussion on the practicality, predicaments and/or implications that might ensue if they were to be implemented in China.

Thank you.

Author Response

Response to Reviewer 1 Comments

Paper ID:ijerph-1742174

Dear Editors and Reviewers:

First of all, we sincerely thank you for your time and constructive comments on our manuscript entitled “The Impact of Digital Economy on the Efficiency of Green Financial Investment in China's Provinces” (ID: ijerph-1742174). Those comments are all valuable and very helpful for revising and improving our paper, as well as the important guiding significance to our researches. We have studied these comments carefully and have made correction which we hope meet with approval. Revised portion are marked in red in the paper. The main corrections in the paper and the point-to-point responses to the editor’s comments are as following:

Reviewer 1

Comments and Suggestions for Authors

Point 1: Some unclear/ misleading sentences-consider revising(e.g., the abstract tests the impact efficiency of digital economy on green financial investment in China by static panel OLS threshold model method….) and many others.

Response 1: Thank you for your opinion, we have revised the abstract part of the article.we increase the digital background of global green finance in the abstract.And we replace the word "text" with "discuss” in abstract . Please check the revised version using the "Track Changes" function in Microsoft Word.

Point 2: Instances of unsubstantiated claims."The digital economy can significantly improve China’s green financial efficiency as a whole…." It could have been better if further elaborations are made in the relevant sections and examples are provided(not necessarily in the abstract itself) . But drawing from what is being claimed in the abstract would enhance readability. I noticed that the authors did make some references to, e.g., improve the efficiency of market operation, etc. But they were all mentioned in passing without making further details within the context of China which I believe that there are a lot of interesting things to tell about.

Response 2: Thank you for your suggestion, We have made the following amendments to your questions.first ,We further explain how the digital economy can improve the efficiency of China's green finance, and illustrate the relationship between the two by using specific examples in China.

Second,Through the introduction and the support of quantitative research results,we can demonstrate “the digital economy can significantly improve China’s green financial efficiency”

Third,in the introduction, specific data of China are added to illustrate that the digital economy is promoting the development of green finance.The modification is as follows.

China's information and communication technology is changing from "following" and "running in parallel" to "leading". The scale of digital consumer market ranks first in the world, and the scale of Chinese netizens ranks first in the world for 13 consecutive years, reaching 1.011 billion in June 2021. The huge scale of netizens promotes the rapid development of digital green finance industry.

Please check the revised version using the "Track Changes" function in Microsoft Word.

Point 3: The literature review could have been better. Rather than merely presenting definitions concepts etc. that are quite straightforward and rather simplistic for this kind of study, the authors may want to consider making some analysis of the previous study of similar nature to critically describe the issues at hand.What is it the main issues that are happening in other contexts and why are they happening or not happening in China.How is China different or similar. Please high The light gaps that would further justify the need for further research. (Literature review gives more analysis, critical thinking, and what gaps have led to our further research)

Response 3: As suggested by the expert, We added more comments on the literature, sorted out the logical structure in addition to related concepts, and summarized the literature from three aspects: concept, index system and existing problems to form a summary.

We find that there are few articles about digital economy directly affecting the efficiency of green finance, so this paper selects the contents related to this article from digital economy and green finance literature for reference.

Please check the revised version using the "Track Changes" function in Microsoft Word.

Point 4: How many provinces were really analyzed? All? The closest information that could be deciphered was".…due to the lack 317 of data in some provinces the lack of data in some years is replaced by the average value 318 of nearly five years "Kindly specify which provinces were analyzed and why (why not) they were (not) analyzed

Response 4: Thank you for your suggestion.We explained why we used the data of these 30 provinces, and the specific amendments are as follows.

”This paper selects the data of digital economy indicators of 30 provinces in China (except Tibet, Hong Kong, Macao and Taiwan) from 2012 to 2018. Due to the lack of digital economy data in Tibet, this paper will not include it in the research object for the time being. Some data of other provinces after 2018 are missing and have not been made public yet. Therefore, this paper selects the provincial data up to 2018, and a small amount of missing data is filled with hot cards.”

Please check the revised version using the "Track Changes" function in Microsoft Word.

Point 5:Please justify the use of the OLS model and threshold model. Although the authors did mention that they are used to deeply study the impact of digital economy on provincial green finance, but little is know why not other models. I believe there is a need to provide some strong justifications to this so that readers might understand the plausibility of engaging such models over the others.

Response 5: Thank you for your suggestion. We explain why OLS model and threshold model are used, and the explanation in the original text is as follows.

“There is little research on the relationship between digital economy and green finance. After studying the relevant literature, our data type is set as the panel data of 30 provinces in China. Therefore, choosing the static OLS model for regression analysis can well reflect and describe whether the relationship between digital economy and green finance is significant, and meet the basic assumptions of this paper. At the same time, we found that the previous empirical analysis of the digital economy's influence factors on green finance is based on the hidden premise, that is, there is no difference in the factor endowments of all regions in China. In fact, the digital economy has to be restricted and influenced by many factors if it is to give full play to its innovative features. Therefore, using the threshold model, considering the influence of external environmental factors, adding threshold variables, and fitting the relationship between digital economy and green finance again, the results of data generation are more scientific, and specific problems can be found. These two methods are in line with the research assumption of this paper, so this paper analyzes the relationship between them and the two threshold variables.”

 Please check the revised version using the "Track Changes" function in Microsoft Word.

Point 6:  Quite a number of instances that the facts and claims were made without any specific citations and references. Indeed some citations and references are not properly cited and referenced e.g.L.P.Rai(2000)Bai Peiwen and Zhang Yun(2021)and Yuan Huaxi and others(2019) . Only surnames are required. Please improve on these by using the approved referencing convention of this journal.

Response 6: Thank you for your suggestion, and we have revised the Literature Review and Research Assumptions section of the article in accordance with this opinion. We have revised the format of references according to the requirements, some of which are as follows, and the rest are revised accordingly. Please check the revised version using the "Track Changes" function in Microsoft Word.

Rai et al.[2]put forward the calculation method of social informatization index earlier

Bai et al.[11] think that the digital economy can stimulate economic growth,

Yuan et al. [20] think that financial support is helpful to the development of green finance from the aspects of capital support, enterprise transformation efficiency, resource allocation and technological innovation .

Point 7:Many recommendations posed in the final part of the paper were not well-thought-of they were just mentioned in passing without much discussion on the practicality predicaments and/or implications that might ensue if they were to be implemented in China.

Response7: Thank you for your suggestion,We revised our policy suggestions.Policy suggestions is put forward according to the previous conclusion. We also put forward scientific and feasible countermeasures and suggestions for China's current predicament.The specific modifications are as follows.

“Accelerate the construction of digital infrastructure, and narrow the development imbalance of green finance regions.”

“Studying the relationship between digital economy and green finance and creating a new format of green digital industry can better help our country to achieve the goal of sustainable development.It is also an important starting point for the financial industry to help supply-side structural reform and better serve the construction of ecological civilization [31].And at the same time digital economy help China achieve the goal of sustainable development, and provide reference suggestions for the regional imbalance of green finance in other countries around the world.”

“Improve the utilization efficiency of digital economy and form a new format of green financial industry.”

“The green financial system should establish a unified background support sharing system for centralized development, operation and maintenance, and effectively reduce the operating and management costs of financial institutions. Using data interconnection technology, a global standard green financial rating system will be established, and green rating standards and methods will be determined to form a new format for the development of green financial industry.”

 ”Improve the development and construction of the green financial system and strengthen the role of audit supervision.”

Therefore, while actively playing the leading role of the market, the government should also pay attention to the role of audit supervision, create a fair and just green financial environment, and increase the development and construction of the green financial system, so as to achieve the three major goals of reducing the green investment cost, increasing the investment cost of polluting projects and improving the social responsibility awareness of enterprises and consumers, and cultivate relevant basic conditions for the rapid and innovative development of green finance.Thank you again for your constructive comments that help us a lot to improve the paper.

Reviewer 2 Report

Dear authors, thank you for your proposal. I consider this research involves two very important topic of our times: digital economy and Green Financial Investment.

However, I have serious reserves about the conceptual structure of the paper, considering that the relationship between the theoretical concepts supporting the research (the digital economy and the green financial investiment) are not adequately supported in theory, nor in pratice, nor even in conceptual terms.

It is natural that the digital transformation of businesses push companies to develop sounder business models. An currently, it is not usual that the new business models are developed without promoting sustainable practices, in line with overall stakeholders' preferences. Assuming this, it is noatural that the development of Digital Transformation shows as an important implication, the development of Green Financial Investment.

In addition, one may assume that younger entrepreneurs tend to develop cheaper technological solutions, and green projects that are financed by different means tend to be supported by digital transformation strategies.

Hsving said this, I consider that:

1- Introduction must be more focused on the true intentions of this paper (it is somehow vague);

2- Literature review must be restructured, considering authors from other geographies that China/Asia, which represent presently more than 95% of total authors involved;

3- Authors must avoid political citations, unless strictly necessary;

4- The theoretical conceptul model must be reviewed and clearly presented, showing a clear line of thought caapble to support authors' vision and recognized in any other country;

Results and Conclusions must be revised accordingly.

Notwithstanding,    

Author Response

Response to Reviewer 2 Comments

Paper ID:ijerph-1742174

Dear Editors and Reviewers:

First of all, we sincerely thank you for your time and constructive comments on our manuscript entitled “The Impact of Digital Economy on the Efficiency of Green Financial Investment in China's Provinces” (ID: ijerph-1742174). Those comments are all valuable and very helpful for revising and improving our paper, as well as the important guiding significance to our researches. We have studied these comments carefully and have made correction which we hope meet with approval. Revised portion are marked in red in the paper. The main corrections in the paper and the point-to-point responses to the editor’s comments are as following:

Reviewer 2

Comments and Suggestions for Authors

Point 1: introduction must be more focused on the true intentions of this paper (it is somehow vague)

Response 1: Thank you for your opinion, we have revised the introduction part of the article.

The first paragraph of the introduction first adds an international perspective to illustrate that the digital economy is promoting the development of traditional financial industry to green finance. At present, China is also facing the problem of environmental pollution, and the emergence of digital economy has provided us with new ideas for the development of green finance. It is conducive to promoting the high-quality growth of economic investment.

The second paragraph of the introduction states that the digital economy is helpful to the development of green finance, and we have found some problems in the research.

In the third paragraph of the introduction, aiming at the above problems, this paper aims to explore whether the digital economy can effectively improve the investment efficiency of green finance and realize the high-quality development of China's economy.

Please check the revised version using the "Track Changes" function in Microsoft Word.

Point 2: Literature review must be restructured, considering authors from other geographies that China/Asia.which represent presently more than 95% of total authors involved:

Response 2: As suggested by the expert, We have added some foreign literature, sorted out the logical structure in addition to related concepts, and summarized the literature from three aspects: concept, index system and existing problems to form a summary. The newly added literature is as follows.Please check the revised version using the "Track Changes" function in Microsoft Word.

[4] Olga Fokina,Sergey Barinov. Marketing concepts of customer experience in digital economy[J]. E3S Web of Conferences,2019,135.

[5] Puchkova Natalia. Business in the digital economy: russian and foreign experience[J]. IOP Conference Series: Materials Science and Engineering,2019,667.

[6] A U Mentsiev,E R Guzueva,S M Yunaeva,M V Engel,M V Abubakarov. Blockchain as a technology for the transition to a new digital economy[J]. Journal of Physics: Conference Series,2019,1399(3).

[7] Philip Davies,Irene Ng. Moving towards the Incomplete: A Research Agenda for the Development of Future Products in the Digital Economy[J]. Procedia Manufacturing,2015,3(C).

[8] Lyapuntsova, E., Y. Belozerova, I. Drozdova, and G. Afanas. (2018) “Entrepreneurial Risks in the Realities of the Digital Economy,” MATEC Web of Conferences 251: 1–6.

[13]Salazar,J.Environmental Finance:Linking Two World[Z].Presented at a Workshop on Finance Innovations for Biodiversity Bratislava,1998,(1),2-18.

[14]Cowan, E., 1999. Topical issues in environmental finance. In: Research Paper was Commissioned by the Asia Branch of the Canadian International Development Agency (CIDA), 1, pp. 1–20.  

[15]ROB GRAY. OF MESSINESS, SYSTEMS AND SUSTAINABILITY: TOWARDS A MORE SOCIAL AND ENVIRONMENTAL FINANCE AND ACCOUNTING[J]. The British Accounting Review,2002,34(4).

[16]Astrid Juliane Salzmann.The integration of sustainability intothe theory and practice of financean overview of the state of the artand outline of future developments. [J].Journal of business economics,2013,(6),555-576.

[17]Soina Labatt and R. R. White,.Environmental Finance:A Guide to Environmental Risk Assessment and Financial Products[J].Advances in Cryogenic Engineering,2002,(8),405-409.

Point 3: Authors must avoid political citations, unless strictly necessary;

Response 3: As suggested by the expert,We delete all the sentences in the introduction and conclusion articles that involve political quotations.Please check the revised version using the "Track Changes" function in Microsoft Word.

Point 4: The theoretical conceptual model must be reviewed and clearly presented showing a clear line of thought capable to support author’s vision and recognized in any other country: Results and Conclusions must be revised accordingly notwithstanding.

Response 4: Thank you for your opinion, We sorted out the research methods, divided the conclusion into separate sections, and drew corresponding policies according to the conclusions. The modification of the research method is as follows

“There is little research on the relationship between digital economy and green finance. After studying the relevant literature, our data type is set as the panel data of 30 provinces in China. Therefore, choosing the static OLS model for regression analysis can well reflect and describe whether the relationship between digital economy and green finance is significant, and meet the basic assumptions of this paper. At the same time, we found that the previous empirical analysis of the digital economy's influence factors on green finance is based on the hidden premise, that is, there is no difference in the factor endowments of all regions in China. In fact, the digital economy has to be restricted and influenced by many factors if it is to give full play to its innovative features. Therefore, using the threshold model, considering the influence of external environmental factors, adding threshold variables, and fitting the relationship between digital economy and green finance again, the results of data generation are more scientific, and specific problems can be found. These two methods are in line with the research assumption of this paper, so this paper analyzes the relationship between them and the two threshold variables. On the one hand, the digital economy plays an increasingly important role in China's financial industry and economic growth, showing strong vitality and becoming one of the important forces to promote China's economic development . On the other hand, green finance is an important driving force to realize "green development", which requires the response and support of green finance [22].”

Please check the revised version using the "Track Changes" function in Microsoft Word.

Thank you again for your constructive comments that help us a lot to improve the paper

Reviewer 3 Report

Digital economy is an emerging economic mode, which has become a rapid growth pole of economic development. At the same time, digital economy is a green industry, which not only promotes the rapid development of regional economy, but also avoids the damage to the regional ecological environment. It is of great significance to formulate practical support policies for the development of digital economy. This research has important practical value and application significance. However, the logical organization is confusing, the manuscript needs to be greatly improved, and some major comments are as follows.

  1. The abstract need to be improved. It is necessary to supplement the highly summarized research background and significance, including reference for global digital economy development.
  2. There are many nonstandard expressions in the manuscript, such as our country, domestic research, which are seriously inconsistent with the internationality of IJERPH.
  3. The whole introduction session only focuses on digital economy development of China. It lacks the description of the global economic development trend and international green economic system. This expression mode is just as a paper published in Chinese journals. The same problem also exists in the Literature Review session.
  4. The manuscript emphasized that green finance is the key to achieving the goal of carbon neutrality. However, the three core research assumptions are not involving the promotion of green economy development on carbon neutralization.
  5. Similarly, the manuscript also emphasized that with the pandemic of the global epidemic, the digital economy become one of the important forces to hedge the impact of the epidemic and promote the gradual economic recovery of China. However, the three core research assumptions are also not involving the impact of digital economy on economic recovery in the post epidemic period.
  6. The logical organization is confusing, especially the research assumptions session. It is suggested to reorganize according to the standard framework of IJERPH.
  7. The quantitative analysis of this research cannot support the policy recommendations. It should put forward practical policy suggestions closely around the research conclusions. For example, according to the threshold effect of digital economy on green finance, it could formulate the differentiated digital economy development countermeasures for each province.

Author Response

Response to Reviewer 3 Comments

Paper ID:ijerph-1742174

Dear Editors and Reviewers:

First of all, we sincerely thank you for your time and constructive comments on our manuscript entitled “The Impact of Digital Economy on the Efficiency of Green Financial Investment in China's Provinces” (ID: ijerph-1742174). Those comments are all valuable and very helpful for revising and improving our paper, as well as the important guiding significance to our researches. We have studied these comments carefully and have made correction which we hope meet with approval. Revised portion are marked in red in the paper. The main corrections in the paper and the point-to-point responses to the editor’s comments are as following:

Reviewer 3

Comments and Suggestions for Authors

Point 1: The abstract need to be improved. It is necessary to supplement the highly summarized research background and significance, including reference for global digital economy development

Response 1: Thank you for your opinion, we have revised the abstract part of the article.we increase the digital background of global green finance in the abstract.And we replace the word "text" with "discuss”in abstract . The specific modifications are as follows.Please check the revised version using the "Track Changes" function in Microsoft Word.

“The global digital operation of finance has accelerated, and the transformation and upgrading of financial industry has been fully empowered by digital technology, which has promoted the development of traditional financial industry to green finance. Accelerating the pace of China's entry into the digital economy era has given the green financial industry new opportunities in the digital transformation. “

Point 2: There are many nonstandard expressions in the manuscript such as our country. domestic research. which are seriously inconsistent with the internationality of IJERPH.

Response 2: Thank you for your suggestion, We make changes to this problem.For example, in the introduction, the nonstandard expression "our country" is completely replaced by "China" .Please check the revised version using the "Track Changes" function in Microsoft Word.

Point 3: The whole introduction session only focuses on digital economy development of China. It lacks the description of the global economic development trend and international green economic system. This expression mode is just as a paper published in Chinese journals. The same problem also exists in the Literature Review session.

Response 3: As suggested by the expert, We have added the contents of the global economic development trend in the abstract, introduction and references, and added the research on green financial system.The specific modifications are as follows.

“In order to solve the limitations of the existing indicators, some scholars further build an evaluation system to measure the comprehensive level of green financial development. Referring to the composite system of green finance development,Put forward an index system to measure the development of China's green finance from five aspects: green credit, securities, insurance, investment and carbon finance. After putting forward the index system of green finance, people pay more attention to the construction between green finance and regional high-quality development. At the same time, international scholars also put forward that the development of green finance needs to rely on the help of society and enterprises to establish the awareness of green finance. “

Please check the revised version using the "Track Changes" function in Microsoft Word.

Point 4: The manuscript emphasized that green finance is the key to achieving the goal of carbon neutrality. However three core research assumptions are  not involving the promotion of green economy development on carbon neutralization

Response 4: Thank you for your suggestion.After research and discussion, although there are many researches on carbon neutralization in related fields. However, this paper is not directly related to it, but just an extension of the research field. In order to prevent ambiguity in reading, the content of carbon neutrality in the conclusion and green finance literature was deleted.Please check the revised version using the "Track Changes" function in Microsoft Word.

Point 5:Similarly the manuscript also emphasized that with the pandemic of the alobaepidemic.the digital economy become one of the important forces to h edge the impact of the epidemic and promote the gradual economic recovery of China. However the three core research assumptions are also not into ving  the impact of digital economy on economic recovery in the post epidemic period

Response 5: Thank you for your suggestion, we have made changes to this problem.After research and discussion, because the data time in this paper is not related to the epidemic situation, the background of the epidemic situation is deleted to prevent ambiguity.Please check the revised version using the "Track Changes" function in Microsoft Word.

Point 6: The logical organization is confusing especially in the research assumptions session. It is suggested to reorganize according to the standard framework of IJERPH.

Response 6: Thank you for your suggestion, and we have revised the Research Assumptions section of the article. Adjust the framework of research assumptions to meet the requirements of journals.Please check the revised version using the "Track Changes" function in Microsoft Word.

Point 7:The quantitative analysis of this research cannot support the policy recommendations. It should put forward practical policy suggestions closely around the research conclusions. For example, according to the threshold effect of digital economy on green finance,  it could formulate the differentiated digital economy development countermeasures for each province

Response7: Thank you for your suggestion, we have revised the policy suggestions section of the article. Part of the policy recommendations of the conclusion are carefully combined according to the quantitative results, such as regional imbalance, insufficient audit supervision and other issues. In the conclusion, some pertinent policy suggestions are put forward, and the experience of China can be used as a reference for the whole world.The specific modifications are as follows .Please check the revised version using the "Track Changes" function in Microsoft Word.

 ”6.1 Accelerate the construction of digital infrastructure, and narrow the development imbalance of green finance regions.”

”The empirical analysis of the influence factors of digital economy on green finance shows that the digital economy is restricted and influenced by many factors if it wants to play its innovative features, so there is an imbalance of regional factors among different provinces in China.”

“6.2 Improve the utilization efficiency of digital economy and form a new format of green financial industry.”

“In the previous research results, it is found that there is a double threshold phenomenon in regional industry scale, and the application efficiency of digital economy in green finance industry is not high enough, which leads to the industry scale gap between different regions. Digital economy and technology help green finance funds to be specially used in the fields of energy conservation, environmental protection and sustainable development, so as to realize the environmental friendly purpose of green finance.”

“6.3 Improve the development and construction of the green financial system and strengthen the role of audit supervision.”

“However, how to better solve the environmental problems and realize the green sustainable development, the market alone can't completely solve the externality problem. We must correct the problem that externality can't be internalized under the market price condition through policy guidance and system construction.”

Thank you again for your constructive comments that help us a lot to improve the paper.

Reviewer 4 Report

The authors of the article formulated three hypotheses: "HYP. 1 The digital economy will improve the efficiency of green financial resources in China's provinces under the premise that other influencing factors are controlled; HYP. 2 Under the premise that other influencing factors are controlled, there is a threshold effect of regional industry size on the impact of digital economy on green financial efficiency; Hyp. 3 Under the premise that other influencing factors are controlled, there is a threshold effect of green financial audit support for the impact of digital economy on green financial efficiency in China's provinces".

The authors claim that: "the digital economy can significantly improve China's green financial efficiency as a whole, and there is a dual threshold effect with regional industry size as the threshold variable and a single threshold with green financial audit support as the threshold variable. The results show that the development of digital economy improves the investment efficiency of green finance in China's provinces as a whole, anddigital economy can improve the financing efficiency of green finance ". The authors point to the possibilities of big data and blockchaine technologies "to establish an information-sharing mechanism through digital service platforms to realize the interconnection of resources, markets, technology and capital", but also emphasize the importance of other factors in the development of a green economy as a result of green investments and their financing.

The given hypotheses were verified on the basis of data from various regional indicators concerning the digital economy and the green economy. The authors assumed that the goal would be achieved by checking whether "provincial panel data, the impact efficiency of digital economy on green financial investment in China by static panel OLS and threshold model method, and constructs threshold model with regional industry scale and green financial audit as threshold variables, in order to analyze the nonlinear characteristics of digital economy and green financial efficiency ". Finding and confirming this relationship allowed the authors to confirm (not reject the accepted hypotheses).

In the opinion of the reviewer, the adopted logic is correct. Of course, other approaches are also possible. The assumed hypotheses are quite obvious, and given the scale of the Chinese economy, it would be difficult to expect them to be unconfirmed. At the same time, the example of China shows that efficiency is achieved with a certain scale of operation. This is not possible in smaller economies. This confirms that the digital economy has a positive impact on the development of the economy, but requires huge investments and is extremely risky (the blocchain crash in 2022). However, it can be concluded that even at the level of ordinary data processing, a certain scale of operation is needed.

Comments:

In the article, it is worth making sure that the requirements of the research process are precisely defined: defining the problem (it is not entirely clear what the relationship will be investigated in what way), defining the purpose of the research, defining the methods (this can be found), defining the contribution to the development of science and practice, perhaps methodology . It is worth referring to this more clearly in the final discussion of the article.

The article end it with recommendations for the Chinese economy. However, it is worthwhile, perhaps in the form of speculation, to indicate more general indicators of the digital economy in the global economy. China has many specific solutions, for example in the context of measures of economic development. It is easier to do in such a situation. It is more difficult when the systems are dispersed and connected only by real nonlinear systems.

Unfortunately, a list of the used literature is unacceptable. Only the achievements of Chinese authors dominate. The necessary condition here is to refer to world literature especially in the context of formulating hypotheses and the very discussion of the results. Otherwise, the potential reception of this article will only be local and useful only on the basis of the indicated local recommendations.

Author Response

Response to Reviewer 4 Comments

Paper ID:ijerph-1742174

Dear Editors and Reviewers:

First of all, we sincerely thank you for your time and constructive comments on our manuscript entitled “The Impact of Digital Economy on the Efficiency of Green Financial Investment in China's Provinces” (ID: ijerph-1742174). Those comments are all valuable and very helpful for revising and improving our paper, as well as the important guiding significance to our researches. We have studied these comments carefully and have made correction which we hope meet with approval. Revised portion are marked in red in the paper. The main corrections in the paper and the point-to-point responses to the editor’s comments are as following:

Reviewer 4

Comments and Suggestions for Authors

Point :In the article, it is worth making sure that the requirements of the research process are precisely defined: defining the problem (it is not entirely clear what the relationship will be investigated in what way),  defining the purpose of the research defining the methods(this can be found)  defining the contribution to the development of science and practice perhaps methodology. It is worth referring to this more clearly in the final discussion of the article.

The article end it with recommendations for the Chinese economy. However it is worthwhile

perhaps in the form of speculation to indicate more general indicators of the digital economy in the global economy. China has many specific solutions for example in the context of measures of economic development it is easier to do in such a situation.It is more difficult when the systems are dispersed and connected only by real nonlinear systems.

Unfortunately, a list of the used literature is unacceptable. Only the achievements of Chinese authors dominate. The necessary condition here is to refer to world literature especially in the

context of formulating hypotheses and the very discussion of the results.Otherwise, the potential reception of this article will only be local and useful only on the basis of the indicated local recommendations.

Response to paragraph 1 : Thank you for your opinion, we have revised the abstract part of the article.

  1. Method question: The abstract and introduction make it clear that the purpose of this paper is whether the digital economy has an effective impact on the investment efficiency of green finance, and many problems are found in the research to solve this part of the problem.

The analysis of the data results makes it clear that our method is in line with the data we selected. Our data type is set as the panel data of 30 provinces in China, so the static OLS model for regression analysis can well reflect whether the relationship between digital economy and green finance is significant, which meets the basic assumptions of this paper. At the same time, we found that the previous empirical analysis of the digital economy's influence factors on green finance is based on the hidden premise, that is, there is no difference in the factor endowments of all regions in China. In fact, the digital economy has to be restricted and influenced by many factors if it is to give full play to its innovative features. Therefore, using the threshold model, considering the influence of external environmental factors, adding threshold variables, and fitting the relationship between digital economy and green finance again, the results of data generation are more scientific, and specific problems can be found.

  1. The relationship between the results and conclusions has been modified, and relevant policy suggestions are put forward more directly through the research results.
  2. References: More international scholars' research has been added, and they have paid attention to the international perspective. In the introduction, the background of the field of international customs clearance is also put forward.
  3. In the discussion part, this problem is also clarified.

Please check the revised version using the "Track Changes" function in Microsoft Word.

Response to paragraph 2 : We divided the conclusion into separate segments, and then described that the indicators and models made are aimed at "the digital economy has a significant impact on green finance". At the same time, for the policy suggestions, we also incorporated the conclusion into the targeted suggestions of green finance digitalization, such as "Accelerate the construction of digital infrastructure, and narrow the development imbalance of green finance regions,Improve the utilization efficiency of digital economy and form a new format of green financial industry and Improve the development and construction of the green financial system and strengthen the role of audit supervision."Please check the revised version using the "Track Changes" function in Microsoft Word.

Response to paragraph 3 : As suggested by the expert, We have added some foreign literature, sorted out the logical structure in addition to related concepts, and summarized the literature from three aspects: concept, index system and existing problems to form a summary. The newly added literature is as follows.Please check the revised version using the "Track Changes" function in Microsoft Word.

[1] Awan Usama,Shamim Saqib,Khan Zaheer,Zia Najam Ul,Shariq Syed Muhammad,Khan Muhammad Naveed. Big data analytics capability and decision-making: The role of data-driven insight on circular economy performance[J]. Technological Forecasting & Social Change,2021,168.

[4] Olga Fokina,Sergey Barinov. Marketing concepts of customer experience in digital economy[J]. E3S Web of Conferences,2019,135.

[5] Puchkova Natalia. Business in the digital economy: russian and foreign experience[J]. IOP Conference Series: Materials Science and Engineering,2019,667.

[6] A U Mentsiev,E R Guzueva,S M Yunaeva,M V Engel,M V Abubakarov. Blockchain as a technology for the transition to a new digital economy[J]. Journal of Physics: Conference Series,2019,1399(3).

[7] Philip Davies,Irene Ng. Moving towards the Incomplete: A Research Agenda for the Development of Future Products in the Digital Economy[J]. Procedia Manufacturing,2015,3(C).

[8] Lyapuntsova, E., Y. Belozerova, I. Drozdova, and G. Afanas. (2018) “Entrepreneurial Risks in the Realities of the Digital Economy,” MATEC Web of Conferences 251: 1–6.

[13]Salazar,J.Environmental Finance:Linking Two World[Z].Presented at a Workshop on Finance Innovations for Biodiversity Bratislava,1998,(1),2-18.

[14]Cowan, E., 1999. Topical issues in environmental finance. In: Research Paper was Commissioned by the Asia Branch of the Canadian International Development Agency (CIDA), 1, pp. 1–20.  

[15]ROB GRAY. OF MESSINESS, SYSTEMS AND SUSTAINABILITY: TOWARDS A MORE SOCIAL AND ENVIRONMENTAL FINANCE AND ACCOUNTING[J]. The British Accounting Review,2002,34(4).

[16]Astrid Juliane Salzmann.The integration of sustainability intothe theory and practice of financean overview of the state of the artand outline of future developments. [J].Journal of business economics,2013,(6),555-576.

[17]Soina Labatt and R. R. White,.Environmental Finance:A Guide to Environmental Risk Assessment and Financial Products[J].Advances in Cryogenic Engineering,2002,(8),405-409.

Thank you again for your constructive comments that help us a lot to improve the paper.

Reviewer 5 Report

The article is very interesting, but should be restructured.
1) The authors should add references to the introduction, and try to summarize the most relevant aspects.
2) Methods and results should also be separated. Establishing and detailing the methods allows the exhaustive review and replicability of the research. Therefore, it is not possible to go straight to the application.
3) It is necessary to separate conclusion from discussion.
4) In the discussion, the findings of your work should be confronted with other literature, contemporary or classic, to put your work in comparative value.

Author Response

Response to Reviewer 5 Comments

Paper ID:ijerph-1742174

Dear Editors and Reviewers:

First of all, we sincerely thank you for your time and constructive comments on our manuscript entitled “The Impact of Digital Economy on the Efficiency of Green Financial Investment in China's Provinces” (ID: ijerph-1742174). Those comments are all valuable and very helpful for revising and improving our paper, as well as the important guiding significance to our researches. We have studied these comments carefully and have made correction which we hope meet with approval. Revised portion are marked in red in the paper. The main corrections in the paper and the point-to-point responses to the editor’s comments are as following:

Reviewer 5

Comments and Suggestions for Authors

Point 1: The authors should add references to the introduction. and try to summarize the most relevant aspects

Response 1: Thank you for your opinion, In the introduction, we added relevant references and sorted them out. The specific added references are as follows.Please check the revised version using the "Track Changes" function in Microsoft Word.

“Awanetal et al.[1] is believed that the financial industry has strengthened the combination with information technology, and a comprehensive model covering different types of enterprises-digital finance has been launched. It is believed that digital finance overcomes the shortcomings of traditional finance, and greatly improves inclusiveness by providing convenient services and lower barriers to entry. From promoting high-quality economic development; It can also help us effectively promote green technology innovation and become the core means to solve global pollution problems”

[1]Awan Usama,Shamim Saqib,Khan Zaheer,Zia Najam Ul,Shariq Syed Muhammad,Khan Muhammad Naveed. Big data analytics capability and decision-making: The role of data-driven insight on circular economy performance[J]. Technological Forecasting & Social Change,2021,168.

Point 2: Methods and results should also be separated. Establishing and detailing the methods allows the exhaustive review and replicability of the research. Therefore it is not possible to go straight to the application.

Response 2: As suggested by the expert, We write the method and the result separately, and supplement and expand the expression of the method.Please check the revised version using the "Track Changes" function in Microsoft Word.

Point 3: It is necessary to separate conclusion from discussion.

Response 3: As suggested by the expert, We added a discussion part to the article, and separated the results from the discussion.Please check the revised version using the "Track Changes" function in Microsoft Word.

Point 4: In the discussion, the findings of your work should be confronted with other literature contemporary or classic, to put your work in comparative value

Response 4: Thank you for your suggestion.We have added a discussion section to the article.Specific content as follows.Please check the revised version using the "Track Changes" function in Microsoft Word.

“There are some limitations in this study. Future research will make up for the shortcomings of this study and provide reference direction for future research. The first limitation is that the data sample only reports the data of 30 provinces in China, and because some data are not public, the data research can be further improved to improve the accuracy of the conclusion. In addition, as a developing country, China is different from other developed countries and other regions in the impact of digital economy on green finance, including Europe, North America and Oceania. Therefore, our future research needs to improve its universality in the world. Another limitation is that the research object of this paper is the data of 30 different provinces in China. It is necessary to establish a unified index system, which cannot be completely accurate to digital economy and green finance. In future research, with the publication of data and the development of the industry, the accuracy of the index system will be continuously improved. There are still differences between the international index system and the Chinese index system. The Chinese index system selected in this paper may not be suitable for international universality. In the next research, we should also establish an index system that conforms to the global norms.

After combing the relevant literature, it is found that few authors directly discuss the impact of digital economy on green finance. This paper directly discusses whether digital economy can improve the investment efficiency of green finance, and fills in the research in this related field. With the help of digital economy technology, we can realize the high-quality development of green finance, which is suitable for China's green finance industry. In the research, we also found the deficiency of the influence of digital economy on green financial development investment, and gave corresponding policy suggestions, which provided reference for scholars in China and other regions.”

Thank you again for your constructive comments that help us a lot to improve the paper.

Round 2

Reviewer 2 Report

Thank you for your new version of the paper, that I consider answers partially to previous reviewers' requests.

I consider that with the changes made, the paper improved both in terms of clarity and support of the ideas.

I am still not very convinced about the soundness of the methodology or the rationale underlying your research topic. I consider that the paper still needs to be improved in terms of the clear relationship that you want to highligth concerning the impact of Digital Economy on the Efficiency of Green Finance Investment in China and related economic development.

Literature review is clear about the specific topics of Digital Economy and Green Finance, but it lacks in developing the relationship between both issues and related economic development. If the relationship between Digital Economy and Economic Development seems clear, there is no direct consideration about the relationship between these and Green Finance. And this is not an easy consideration, as authors acknowledge in page 7/22, lines 342-343, at the same time they acnowledge this is a complex mechanism (see p. 8/22, lines 379-381).

Hypothesis 1 and 2 relate to almost the same idea, and they do not allow to foresee a specific relationship between independent and dependent variables. Hypothesis 3 could be interesting, but I seriously doubt of its coherence, giving the fact that the relationship was not clearly defined previously (nor in literature, nor in the paper).

Despite these observations, I consider that the potential of the paper remains very interesting, namely if authors are capable to develop a clear relationship between Green Financial Indicators (as shown in table 3) and the Digital Economy Indicators (Tables 1 and 2).

A very positive point of the new version is the consideration of the limitations, as previously requested.

From my point of view this research may provide a very interesting result, but authors need to be able to clearly identify the relationship between Digital Economy (and they need to clearly define what they consider by digital economy, and whether they are not talking about digital transformation of the economy) and Green Finance. 

Author Response

Response to Reviewer 2 Comments

Paper ID:ijerph-1742174

Dear Editors and Reviewers:

First of all, we sincerely thank you for your time and constructive comments on our manuscript entitled “The Impact of Digital Economy on the Efficiency of Green Financial Investment in China's Provinces” (ID: ijerph-1742174). Those comments are all valuable and very helpful for revising and improving our paper, as well as the important guiding significance to our researches. We have studied these comments carefully and have made correction which we hope meet with approval. Revised portion are marked in red in the paper. The main corrections in the paper and the point-to-point responses to the editor’s comments are as following:

Reviewer 1

Comments and Suggestions for Authors

Point 1: I am still not very convinced about the soundness of the methodology or the rationale underlying your research topic. I consider that the paper still needs to be improved in terms of the clear relationship that you want to highligth concerning the impact of Digital Economy on the Efficiency of Green Finance Investment in China andrelated economic development.

Literature review is clear about the specific topics of Digital Economy and Green Finance,but it lacks in developing the relationship between both issues and related economic development. If the relationship between Digital Economy and Economic Development seems clear, there is no direct consideration about the relationship between these and Green Finance. And this is not an easy consideration,as authors acknowledge in page 7/22lines 342-343,at the same time they acnowiedge this is a complex mechanism(seep.8/22lines 379-381).

Response 1: As suggested by the expert, In the reference section, we added the section "Research on the Effect of Digital Economy on Green Finance Efficiency" to study the relationship between digital economy and green finance.Please check the revised version using the "Track Changes" function in Microsoft Word.

Point 2: Hypothesis 1 and 2relate to almost the same idea, and they do not allow to foresee a specific relationship between independent and dependent variables. Hypothesis 3 could be interesting. but i seriously doubt of its coherence,giving the fact that the relationship was not clearly defined previously (nor in literature nor in the paper).

Response 2: Thank you for your suggestion, We further elaborate hypothesis 3 to make it more consistent with hypothesis 2, and add references to hypothesis 3 to illustrate why we should make this hypothesis.The specific contents are as follows.Please check the revised version using the "Track Changes" function in Microsoft Word.

“Two external environments are selected respectively: regional industry scale and green financial audit. Because the larger the industry scale, the better the digital facilities of green finance are generally built. The choice of green financial audit is because, while the scale is expanding, the traditional audit mode can't meet the development of green finance, which is lagging behind and passive. Therefore, only by joining the external condition of green financial audit, standardizing the operation behavior of market participants as much as possible, strengthening the legal construction of financial industry and supervising the effectiveness of policy implementation can we prevent and control the systemic risks of green finance to the maximum extent.These two methods are in line with the research assumption of this paper, so this paper analyzes the relationship between them and the two threshold variables.”

[33]Yang Chao, Wang Tianyu. Financial audit promotes the development of green finance [J]. China Finance, 2018(20):47-48.

Point 3: namely if authors are capable to develop a clear relationship between Green Financial Indicators(as shown in table 3) and the Digital Economy Indicators(Tables 1and2)

Response 3: As suggested by the expert, We added the description of 3.3.2 green financial indicators, and expounded the construction of green financial indicators in the digital age. The specific additions are as follows.Please check the revised version using the "Track Changes" function in Microsoft Word.

“Technological innovation in the era of digital economy is conducive to the development of green finance. At present, the index construction of green finance is mainly embodied in four dimensions, namely, green credit, green investment, green insurance and government support. Four specific secondary indicators are selected to describe the development of green finance in China's provinces in the era of digital economy.”

Thank you again for your constructive comments that help us a lot to improve the paper.

Reviewer 3 Report

Digital economy is an emerging economic mode, which has become a rapid growth pole of economic development. This research has important practical value and application significance. In the revised manuscript, the author has made carefully revised and improvement. There are still some minor modification suggestions for improving the article.

1.      In the last part of Introduction, there are still some inappropriate statements about the research limitation.

2.      Digital economy is a green industry, which plays an important role in reducing energy consumption and protecting the environment. However, the research significance is not about the carbon emission reduction effect of digital economy. It is no need special emphasis.

3.      There are repeated and crossed titles. For example: 2.3. Research Methods and Research Assumptions, 3. Research Assumptions. And some content and title are inconsistent. The framework could be summarized as: 1. Introduction, 2. Literature Review, 3. Research Methods and Research Assumptions, 4. Result, 5. Discussion, 6. Conclusion.

Author Response

Response to Reviewer 3 Comments

Paper ID:ijerph-1742174

Dear Editors and Reviewers:

First of all, we sincerely thank you for your time and constructive comments on our manuscript entitled “The Impact of Digital Economy on the Efficiency of Green Financial Investment in China's Provinces” (ID: ijerph-1742174). Those comments are all valuable and very helpful for revising and improving our paper, as well as the important guiding significance to our researches. We have studied these comments carefully and have made correction which we hope meet with approval. Revised portion are marked in red in the paper. The main corrections in the paper and the point-to-point responses to the editor’s comments are as following:

Reviewer 3

Comments and Suggestions for Authors

Point 1: In the last part of Introduction.there are still some inappropriate statements about the research limitation.

Response 1: Thank you for your suggestion. We deleted the expression on the limitations of research in the third paragraph of the introduction, and added 2.3 literature on the impact of digital economy on green finance to the literature review, and described the limitations of the article in the last paragraph of this part.Please check the revised version using the "Track Changes" function in Microsoft Word.

Point 2: Digital economy is a green industry which plays an important role in reducing energy consumption and protecting the environment However the research significance is not about the carbon emission reduction effect of digital economy.It is no need special emphasis.

Response 2: Thank you for your suggestion. We have deleted the statements similar to carbon emissions. For example, the first paragraph of the introduction deleted the original carbon emissions data of China.Please check the rest revised version using the "Track Changes" function in Microsoft Word.

Point 3: There are repeated and crossed fitles For example:2.3. Research Methods and Research assumptions. 3.Research Assumptions.And some content and title are inconsistent The framework could be summarized as 1 Introduction 2 literature Review 3 Research Methods and Research Assumptions, 4.Result ,5. Discussion,6.Conclusion.

Response 3: As suggested by the expert, According to your suggestion, we combined 2.3 in the literature review in the previous edition with the original third part to form the part of Research Methods and Research Assumptions. We think this is more in line with the title requirements of the article. As the last part of this article is policy recommendations, the titles of our parts 4, 5 and 6 have not changed.Please check the revised version using the "Track Changes" function in Microsoft Word.

Reviewer 5 Report

Dear authors, only 2 of the requested issues still seem to me to be unresolved.

1) Regarding the separation between methods and results. At what number do the "Research Assumptions" end and the results begin? It seems to be just a problem of re-numbering the items.

2) I cannot see how the authors claim to compare contemporary and classical literature. If section 4 Discussion, it does not incorporate any citation to any reference.

Author Response

Response to Reviewer 5 Comments

Paper ID:ijerph-1742174

Dear Editors and Reviewers:

First of all, we sincerely thank you for your time and constructive comments on our manuscript entitled “The Impact of Digital Economy on the Efficiency of Green Financial Investment in China's Provinces” (ID: ijerph-1742174). Those comments are all valuable and very helpful for revising and improving our paper, as well as the important guiding significance to our researches. We have studied these comments carefully and have made correction which we hope meet with approval. Revised portion are marked in red in the paper. The main corrections in the paper and the point-to-point responses to the editor’s comments are as following:

Reviewer 5

Comments and Suggestions for Authors

Point 1:Regarding the separation between methods and results.At what number do the"Research Assumptions"end and the results begin?It seems to be just a problem of re-numbering the items.

Response 1 : As suggested by the expert, According to your suggestion, we combined 2.3 in the literature review in the previous edition with the original third part to form the part of Research Methods and Research Assumptions. We think this is more in line with the title requirements of the article. As the last part of this article is policy recommendations, the titles of our parts 4, 5 and 6 have not changed.Please check the revised version using the "Track Changes" function in Microsoft Word.

Point 2: I cannot see how the authors claim to compare contemporary and classical literature.If section4 Discussion. it does not incorporate any citation to any reference.

Response 2 : Thank you for your suggestion. We added the literature on the impact of digital economy on green finance in the literature review section, and found its limitations through research, which led to the discussion in the discussion section.Please check the revised version using the "Track Changes" function in Microsoft Word.

[22]Grossman G, Kruger A. Environmental Impacts of the North American Free Trade Agreement[R]. NBER, Working Paper, 1991, (11) :1-57.

[23]He Xingbang. Environmental regulation and the quality of China's economic growth-an empirical analysis based on provincial panel data [J]. Contemporary Economic Science, 2018,40(2):1-10.

[24]Wang Qunyong, Lu Fengzhi. Can environmental regulation boost the high-quality development of China's economy? —— Empirical test based on provincial panel data [J]. Journal of Zhengzhou University (Philosophy and Social Sciences Edition), 2018,51(6):64-70.

[25]Ciocoiu, Carmen Nadia. INTEGRATING DIGITAL ECONOMY AND GREENECONOMY OPPORTUNITIES FOR SUSTAINABLE DEVELOPMENT[J].Theoretical and Empirical Researches in Urban Management. 2011, 6(1): 33-43.

[26]Zheng Xiaoyun, Cassandra, Su Yikun. Research on the coordinated development of green economy and digital economy-an empirical analysis based on the modified coupling model [J]. Price Theory and Practice, 2021,(08):164-167+187.

[27]Xue Wei. Combination of Digital Economy and Green Economy-Application of Internet of Things [J]. Informatization Construction, 2016(3):103

[28]Qian Lihua, Fang Qi, Lu Zhengwei, Research on Synergy between Green Economy and Digital Economy in Stimulating Policy [J]. Southwest Finance, 2020(12):3-13.

Thank you again for your constructive comments that help us a lot to improve the paper.
